# Diversity and Regulation of S-Adenosylmethionine Dependent Methyltransferases in the Anhydrobiotic Midge

**DOI:** 10.3390/insects11090634

**Published:** 2020-09-16

**Authors:** Ruslan Deviatiiarov, Rustam Ayupov, Alexander Laikov, Elena Shagimardanova, Takahiro Kikawada, Oleg Gusev

**Affiliations:** 1Institute of Fundamental Medicine and Biology, Kazan Federal University, Kazan, Tatarstan 420008, Russia; alexander.laikov@yandex.ru (A.L.); rjuka@mail.ru (E.S.); gaijin.ru@gmail.com (O.G.); 2Department of Cell Biology, SUNY Downstate Health Sciences University, Brooklyn, NY 11203, USA; aurusta@mail.ru; 3Anhydrobiosis Research Group, Molecular Biomimetics Research Unit, Institute of Agrobiological Sciences, National Institute of Agriculture and Food Research Organization (NARO), Tsukuba 305-0851, Japan; kikawada@affrc.go.jp; 4RIKEN-KFU Translational Genomics Unit, RIKEN Cluster for Science, Technology and Innovation Hub, RIKEN, Yokohama 230-0045, Japan

**Keywords:** SAM-dependent methyltransferases, PIMT, *Polypedilum vanderplanki*, anhydrobiosis, transcriptomics

## Abstract

**Simple Summary:**

Adaptation to anhydrobiotic conditions of *Polypedilum vanderplanki* at the larval stage is accompanied by specific genome features and related regulatory mechanisms. The unusual diversity of paralogous genes located in gene clusters with a strong dehydration specific response is thought to underlie the desiccation tolerance of the insect. One of the most representative clusters consists of the gene coding protein, L-isoaspartate O-methyltransferases (PIMT), but it remains poorly characterized. In our work, by applying the transcriptomic RNA-seq approach on desiccated-rehydrated larvae, we showed that these genes have significant dissimilarities in their transcriptional activity within the group, but also in comparison with the expression profiles of other defined types of S-adenosylmethionine dependent methyltransferases active in the larvae. We also showed the standard methylation activity of two PIMTs, while the rest of the 12 tested proteins lacked an enzymatic function in normal physiological conditions. These results, together with in silico modelling, determine the heterogeneity of the group in terms of its role in the adaptation to anhydrobiosis.

**Abstract:**

Multiple co-localized paralogs of genes in *Polypedilum vanderplanki’s* genome have strong transcriptional response to dehydration and considered to be a part of adaptation machinery at the larvae stage. One group of such genes represented by L-isoaspartate O-methyltransferases (PIMT). In order to highlight specific role of PIMT paralogization in desiccation tolerance of the larvae we annotated and compared S-adenosylmethionine (SAM) dependent methyltransferases of four insect species. From another side we applied co-expression analysis in desiccation/rehydration time course and showed that PIMT coding genes could be separated into five clusters by expression profile. We found that among *Polypedilum vanderplanki’s* PIMTs only PIMT1 and PIMT2 have enzymatic activity in normal physiological conditions. From in silico analysis of the protein structures we found two highly variable regions outside of the active center, but also amino acid substitutions which may affect SAM stabilization. Overall, in this study we demonstrated features of *Polypedilum vanderplanki’s* PIMT coding paralogs related to different roles in desiccation tolerance of the larvae. Our results also suggest a role of different SAM-methyltransferases in the adaptation, including GSMT, JHAMT, and candidates from other classes, which could be considered in future studies.

## 1. Introduction

The larvae of the sleeping chironomid, *Polypedilum vanderplanki* (Pv), have adapted to survive severe dehydration by entering an anhydrobiosis state [1], a kind of cryptobiosis [2]. According to previous studies, during desiccation, the larvae accumulates trehalose and late embryogenesis abundant proteins (LEA), which form a glassy structure instead of a water pool. Together with heat shock and redox proteins, this structure provides conditions for the safe preservation of cellular components [3]. After sequencing and genome assembly, a new set of key genes involved in the anhydrobiosis and its regulation were uncovered [4]. In comparison with the genomes of other animals, including the closely related species *Polypedilum nubifer*, *P. vanderplanki*’s genome contains a large group of paralogous L-isoaspartate O-methyltransferases (PIMT)-like genes [4]. Such PIMTs belong to group of S-adenosylmethionine (SAM) dependent methyltransferases (MT), responsible for transferring the methyl group to the L-isoaspartate/D-aspartyl residues in the amino acid sequences of numerous cell proteins, for example, from S-adenosyl-l-methionine (SAM) to either the a-COOH group of an l-isoaspartyl residue (isoAsp) or the b-COOH group of a d-aspartyl residue [5]. This reaction is linked to the consequences of the spontaneously occurring degradation reaction of L-Aspartyl (L-Asp) and L-asparaginyl (L-Asn), resulting in the generation of atypical D-Asp and D,L-isoAsp residues, resulting in protein aging, inactivation, and aggregation, even under normal conditions, which could be boosted by desiccation and other abiotic stresses [6,7,8]. The generated isoL-Asp is partially reversed back to L-Asp, because of the activity of the methyltransferases, showing the vital role of the PIMT-mediated enzymatic reaction.

In general, PIMT-coding genes are widespread and are represented by a single copy in prokaryotes, plants, and animals’ genomes [9,10,11]. Surprisingly, the genome of anhydrobiotic midge contains an extra 14 paralogs of *Pimt* arranged in compact gene clusters, called anhydrobiosis-related gene islands (ARIds) [4]. Such ARIds are considered to be a distinctive feature of anhydrobiosis adaptation, and point to the specific trait of the evolution of the larvae’s adaptation mechanisms. Genomic sequencing of a closely related species, *P. nubifer,* which is sensitive to water loss, revealed no ARId-like regions and only a single *Pimt* gene [4].

In this work, we conducted a whole-genome comparative survey of all accessible SAM-dependent methyltransferases in order to specify role of PIMTs and to define the previously unaccounted anhydrobiosis-related genes. The number of paralogs within dipteran species and expression response to desiccation stress were used as markers of the relations to the adaptation. Finally, computational and biochemical approaches revealed differences in functional specialization among *PvPimt* genes, and a significant role in desiccation tolerance among SAM MTases.

## 2. Materials and Methods

### 2.1. Identification of SAM-Dependent Methyltransferases

SAM-dependent methyltransferases coding genes in the fruit fly, *Drosophila melanogaster,* and in two chironomids, *P. vanderplanki* and *P. nubifer,* were identified by generating a GO Slim using OBO-Edit2 program (http://oboedit.org/?page=index), using GO:0008757 (S-adenosylmethionine-dependent methyltransferase activity) as a parent class (Appendix A). For the identification of the non-annotated MT genes in the chironomids, we applied tblastx (search translated nucleotide databases using a translated nucleotide query). The genes from *D. melanogaster* were used as the query for the tblastx search. Selected gene sequences were translated to proteins and were submitted to protein BLAST to verify whether the nearest homolog was known as putative methyltransferase. Gene sequences from *D. melanogaster* in GFF format were downloaded from FlyBase (http://flybase.org/static_pages/downloads/bulkdata7.html). For the chironomids *P. vanderplanki* and *P. nubifer*, sequences were acquired from MidgeBase (http://bertone.nises-f.affrc.go.jp/midgebase/). For the Antarctic chironomid, *Belgica antarctica,* sequences were obtained from the NCBI database (http://www.ncbi.nlm.nih.gov/Traces/wgs/?&val=GAAK01.1&size=50&display=contigs&page=1#list).

### 2.2. Differential Expression Analysis and Clustering

The transcriptomics data were obtained from NCBI SRP070984 [12], mapped with TopHat to show the current genome assembly of *P. vanderplanki*, and counted with HTSeq-count (nonunique none). For the differential expression analysis, edgeR package for R was used. The level of gene expression was compared at different hours after slow desiccation inception (0, 1, 8, 12, 16, 24, 32, 36, 40, and 48 h) and further rehydration (0, 1, 3, 6, 12, 24, and 48 h). All of the time experimental points were duplicated or triplicated, except for 32 and 36 h of desiccation and 1, 6, 12, and 48 h of rehydration. In order to compare the methyltransferase gene response to water loss, we compared the neighbor experimental points (Appendix A) and everything else against the control D00 (Appendix A). The trends of expressions were defined using the self-organizing maps (SOM) package for R. Co-expressed clusters of genes were defined though the WGCNA package (soft threshold 5). For the SOM and WGCNA analyses, we used all of the expressed genes (*n* = 16,400).

### 2.3. Modelling of PIMTs and SAM Interactions

Fourteen models of PIMT proteins were constructed using Phyre2 (http://www.sbg.bio.ic.ac.uk/phyre2/html/page.cgi?id=index) and Robetta (http://robetta.bakerlab.org/) programs. The nearest homologous protein with known crystal structure for *P. vanderplanki*’s PIMTs was 1R18 from *D. melanogaster* (http://doi.org/10.2210/pdb1R18/pdb). Molecular docking between the SAM molecule and the predicted models of the PIMT proteins was performed. The initial position of the molecule was taken from 1R18. Docking was carried out in the program AutoDock, using its application vina.exe (http://autodock.scripps.edu/).

### 2.4. Recombinant Protein Purification

The pPAL7 vector system (Bio-Rad Laboratories, Hercules, CA, USA) was used for the induction of recombinant PIMTs in *E. coli* BL21(DE3). Vectors with *PvPimt* genes were artificially synthesized by GenScript Biotech (Piscataway, NJ, USA). The cells were lysed using the “BugBuster” protein extraction reagent (Merck Millipore, Burlington, MA, USA), centrifuged at 16,000× *g* for 20 min at 4 °C, and 1 or 5 mL of supernatant were used for fast protein liquid chromatography (FPLC) on NGC™ Medium-Pressure Chromatography Systems (Bio-Rad) with 1 or 5 mL of eXact column (Bio-Rad). The purified proteins were stored at −20 °C in a phosphate buffer with 10% glycerol until use. SDS PAGE was used for electrophoretic separation using a “sample buffer” with a composition of ×2 0.1 M Tris HCl, pH 6.8, 4% SDS, 12% β-mercaptoethanol, 20% glycerol, ≈0.0001% BPB, and a “running buffer”, with a composition of 12% Tris, 13.3% aspartic acid, 1% SDS, pH 6.0, in polyacrylamide gel with a composition of 0.3M Tris HCl, pH 7.2, 9% acrylamide, 0.1% SDS, 0.1% APS, and 0.1% TEMED, with a protein amount of 1 µg.

### 2.5. Protein Identification

Purified PIMT recombinant proteins were digested with trypsin and used in LC–MS. The chromatography properties were as follows: column AcclaimPepMap RSLC, ø 75 µm, length 15 cm, C18, 2 µm particle sizes, and 100 Å pore sizes (Thermo Fisher Scientific, Waltham, MA, USA). The precolumn properties were as follows: AcclaimPepMap 100, ø 75 µm, length 2 cm, C18, 3 µm particle sizes, and 100 Å pore sizes. The separation temperature was 40 °C. Solution A consisted of the following: 94.9% H_2_O, 5% acetonitrile, and 0.1% formic acid. Solution B consisted of the following: 94.9% acetonitrile, 5% H_2_O, and 0.1% formic acid. Chromatography was done for 5 min with 2% B (application and desalination in the precolumn at a flow rate of 5 µL/min), 15 min for gradient B from 2% to 50%, 1 min from 50% to 90% for B, 7 min for 90% B, 2 min from 90% to 2% B, and 5 min for 2% B. The flow rate in the column was 0.3 µL/min. The injection volume was 10 µL. The autosampler was taken at 10 °C. The chromatography system used was UltiMate 3000 UHPLC system (Thermo Fisher Scientific, Waltham, MA, USA).

Mass spectrometry properties were the detection of positive charged ions, voltage across the capillary of 1400 V, frequency of 2 Hz, ionization source CaptiveSpray (gas flow 3 L/min, 150 °C), applicable mass range 50–2200 *m*/*z*, and autoMS/MS mode. The MS system was maXis Impact, Bruker (Billerica, MA, USA).

Mass spectrum analysis was performed by DataAnalysis 4.1 software (Bruker). The final protein identification was made in Mascot 2.4.0 (database of proteins based on transcriptomics data; http://www.matrixscience.com/).

### 2.6. Enzymatic Activity Assay

The HPLC for the activity assay was performed as proposed by Furuchi et al. [13]. NBD-DSIP peptide (≈95% purity) was purchased from Biotest Systems, Moscow, Russia. The activity assay was conducted using UltiMate 3000 UHPLC system equipped with Reaction Column Acclaim 120 C18 (Thermo Fisher Scientific, Waltham, MA, USA).

## 3. Results

### 3.1. The Variety and Differential Expression of SAM-Dependent Methyltransferases in Dipteran Insects

We performed a genome-wide search of all SAM-dependent *MTases* in four species, including three chironomids, namely, *P. vanderplanki*, *P. nubifer*, and *B. antarctica,* and the fruit fly, *D. melanogaster*.

For the fruit fly, we found 85 methyltransferases genes with known SAM-dependent activity. We divided them into several groups, depending on the type of reaction or the acceptor type of the methyl residue (Table 1 and Appendix A for details). In the chironomids, we discovered 102, 88, and 65 MTases for *P. vanderplanki*, *P. nubifer*, and *B. antarctica*, respectively. Among all *P. vanderplanki* genes, the group of *PvPimt* was of high interest because of the unusually high number of paralogs (*n* = 15), while only one *Pimt* ortholog was present in all of the other insects. We also found an increased number (*n* = 13) of genes for ubiquinone biosynthesis O-methyltransferases and juvenile hormone acid methyltransferases (*n* = 4) in the *Polypedilum* species, in comparison with the fruit fly or Antarctic midge. Finally, *P. vanderplanki* has an additional gene of Glycine/Sarcosine methyltransferase and variable numbers of DNA/RNA/histone MTases (Table 1).

To generalize the *MTases*’ response to dehydration stress, we applied SOM clustering based on expression profiles during the desiccation–rehydration steps and separated the genes into three groups (Figure 1a): upregulation trend (blue), stable (red), and downregulation trend (turquoise). From these data, we can conclude that most of expressed genes are sensitive to desiccation (≈13.5K) and have positive or negative regulation. According to DE analysis, 8706 genes in total responded to desiccation at some timepoint (FDR < 0.05). Among the *MTases*, there were 67 genes with statistically significant response; interestingly, three of them changed expression within the first hour of larval dehydration. The most drastically regulated gene was DNA adenine-specific methyltransferase *PvDNA-MT3*, which presented a 124-fold decrease in its expression in comparison to the normal state (Appendix A). Another inactivated gene was Glycine/Sarcosine N-methyltransferase *PvGsmt2* with −3.4 FC (Figure 1b). Only one gene, Ubiquinone biosynthesis O-methyltransferase *PvUbmt12*, showed positive regulation with 4.4-fold increased expression (Appendix A). Some other *MTases* with reasonable expression levels are also noted in Figure 1b.

Almost all *PvPimt* genes found to be activated within dehydration (except *PvPimt1*, see Appendix A) had the highest expression rate among the studied methyltransferases at all desiccation–rehydration stages (Figure 1d), and despite a high sequence similarity, tended to have different expression profiles (Figure 1c). Other *MTases* groups were also regulated in a specific manner, for example, among *PvUbmt* genes, only *PvUbmt5* and *PvUbmt13* were classified as upregulated, but had an early and late desiccation stages response. In the case of the *RNA-MTases* group, the most highly expressed genes, *PvRNA-MT10* and *PvRNA-MT6*, showed a late desiccation and early rehydration related response, respectively (Figure 1b). Among the histone methyltransferases classified as upregulated *PvHMT-8*, *PvHMT-15*, *PvHMT-19*, and *PvHMT-20* changed expression level with statistical significance (FDR < 0.05), and if *PvHMT-15* was activated at 8 h, the of the rest genes had a 40–48 h specific activation.

For the accurate classification of *MTases* in Pv, we applied co-expression analysis using the WGCNA package on the entire transcriptomic data accessed from Mazin et al., 2018. The heatmap in Figure 2a represents the topological overlap scores (TOM-based) and reveals two relatively large co-expressed clusters of *MTases* #1 and #4 (see Appendix A for related genes). Notably, the clustered genes in these cases belong to variable classes, including *Pimt*, *Ubmt*, *HMT*, *RNA-*, and *DNA-MTases*. Regarding the PIMT coding genes, they were separated by co-expression analysis into five different clusters (Figure 2b,c and Appendix A). *PvPimt5* and *PvPimt14* were found to have an early desiccation specific expression profile, while the rest of the genes, except *PvPimt1*, had a late (#1) or common (#2, #4) desiccation related response (Figure 2c).

### 3.2. The Structure Prediction of Anhydrobiosis Specific PIMTs

Based on similarity, 15 *PvPimt* gene products of *P. vanderplanki* were assigned to the same class EC 2.1.1.77 of protein-L-isoaspartate (D-aspartate) O-methyltransferases; however, some structural differences were observed. Multiple alignments gave an identity range of 31–74% (Appendix A). Modeling analysis showed that *P. vanderplanki*’s PIMTs overlap the 1R18 methyltransferase of *D. melanogaster,* almost through the whole length, except two regions—from 103 to 113—and a second small region—from 41 to 46 (Figure 3a).

These regions are not related to the active site, suggesting that they are not involved in the catalytic activity. We also obtained the docking energy of the SAM affinity to the protein (Table 2) and the new coordinates of the SAM molecule for each PIMT protein (Figure 3). Figure 3b shows the positions of the SAM molecule in six PIMT structures, which are similar to the SAM position in 1R18. The positions of SAM in other predicted structures were different (Figure 3c,d). Multiple amino acid sequence alignments of these proteins revealed that in PIMT7 and PIMT10, the position of 60(61) is occupied by Pro, and for PIMT 3, 4, 9, and 13 by Ala. In other proteins, this position was occupied by Ser, as in reference 1R18 (Appendix A). His118 was related to the heterocycle stabilization of SAM in reference 1R18—a highly variable position among PIMTs (seven different amino acids).

### 3.3. Mass Spectrometry and Enzymatic Activity of Recombinant PIMTs

All PIMT proteins were collected and purified up to 87–99% (Appendix A) through the NGC fast protein liquid chromatography system, except for PIMT8 and PIMT6.2 (skipped due to a relatively low gene expression). Mass spectrometric analysis confirmed the similarity between recombinant proteins and the expected gene products (*p*-value < 0.05; Appendix A). Surprisingly, most of PIMTs were inactive against a common substrate—NBD-DSIP peptide with isoAsp residue in the middle—except for PIMT1 and another PIMT2 with Km for peptide, which were 360.18 and 1041, respectively (Appendix A).

## 4. Discussion

SAM-dependent methyltransferases include numerous enzymes that use SAM as a methyl donor for DNA, RNA, small molecules, histones, and numerous cell protein methylations [14]. They play essential roles in the metabolism, such as the regulation of the gene expression, in the stabilization and protection of the cell components, and are involved in numerous biosynthesis pathways [14,15]. We compared the number of *SAM MTases* paralogs between insect species with tolerance to severe environmental conditions (*P. vanderplanki* and *B. antarctica*) and control species without such adaptations (*P. nubifer* and *D. melanogaster*) in order to check for duplication events. Some genes have obvious *Polypedilum* genus-related extensions, like *PvUbmt* and *PvJhamt*, while *PvPimt* paralogs is a specific trait of anhydrobiotic midge. In addition, no such duplication events were found for *B. antarctica*.

From bacteria up to plant and animal species, there are one or two PIMT coding genes expressed under environmental or hormonal regulation [10,16]. In our current study, the detailed dehydration time course confirms that all ARId related *PvPimt* genes have desiccation specific expression profiles, except for the *PvPimt1* gene which has an inactivation trend (Figure 2c, and Appendix A for details). Interestingly, genes coding the LEA proteins, which are also known to be involved in seed resistance to dehydration, have a different temporal expression in the larvae of *P. vanderplanki* [17], and showed an early or stable desiccation-specific response in the same way to *PvPimt*. For example, *PvLea5, PvLea9, PvLea12, PvLea15*, and *PvLea26* are co-expressed with cluster #1 *MTases*, which tend to be active during the whole dehydration period, while *PvLea2, PvLea4, PvLea6*, and *PvLea11* upregulated at the beginning, together with *PvPimt5* and *PvPimt14* (data not shown). It is possible that *PvPimt* genes with similar expression profiles may have a different tissue specificity, as it was shown for mice brain and testes [18], plants tissues and seeds [16], or to have a different cellular localization in the cytosol or nucleus [19].

The studied PIMTs have a Rossman fold (Figure 3a) and SAM binding motif (Appendix A), but also show significant changes in the SAM binding orientation (Figure 3c,d) and substitutions of catalysis related amino acids like His118 or Ser60 [11]. In the PIMTs, 5–8 substitution of Ser60 to Ala60 show a lack of catalytic activity. The enzymatic activity assay confirmed the loss of enzymatic functions of most Pv PIMTs in normal physiological conditions, while only PIMT1 and PIMT2 were able to methylate the isoAsp residue. All of the PIMT proteins were not able to methylate peptides with D-aspartate or D-isoaspartate, which are usually unrepairable [6,20]. According to a recent study, the larvae accumulates citrate during desiccation [21], therefore the enzymatic activity of PIMTs might also be affected by pH. An alternative hypothesis proposes PIMTs to be a part of the stress specific signal/metabolic pathways, even without any methyltransferase activity [22,23,24]. There is a similar conclusion, that PIMT may have specific functions in different plant tissues, despite its enzymatic activity [25].

We found that the genes of other methyltransferases, like glycine/sarcosine N-methyltransferase (*PvGsmt2*), histone-lysine N-methyltransferases, ubiquinone biosynthesis O-methyltransferases (*PvUbmt*), RNA-MT, and juvenile hormone acid methyltransferase (*PvJhamt*), are involved in anhydrobiosis adaptation. One *Gsmt* gene of *P. vanderplanki* drastically changed its activity and is comparable to the *PvPimt* gene’s expression rate (Figure 2b). GSMT catalyzes glycine methylation and is involved in the osmotic stress resistance of plants and bacteria species [26]. In *D. melanogaster,* this enzyme is a key regulator of SAM levels, and its overexpression could increase the lifespan of the fruit fly through SAM catabolism [27]. A strong transcriptional response to the desiccation of the *Jhamt* gene supposes the maintenance of the physiological state through juvenile hormone related pathways [28].

Epigenetic transcriptional regulation is a part of anhydrobiosis related events occurring in the larvae after various HMTs are activated. Another gene that is highly expressed in dehydration experiments is RNA cap guanine-N2 methyltransferase (*PvRNA-MT10*). Such an MT is a part of capping enzyme complex, playing a role in the transcription regulation [29].

## 5. Conclusions

Our work proposes new insights into the molecular mechanisms of anhydrobiosis adaptation through the transcriptional regulation of SAM-binding MTases. At this moment, we assume that new *PvPimt* genes are required for dehydration tolerance and have different roles in this process in Pv larvae. The demonstrated case of multiple paralogs is a good model for further PIMT research in order to uncover new functional roles of this protein. In addition, we showed examples of other SAM MTases genes with expression profiles comparable to that of *PvPimt*, and they should therefore be considered in further anhydrobiosis-related studies.

## Figures and Tables

**Figure 1 insects-11-00634-f001:**
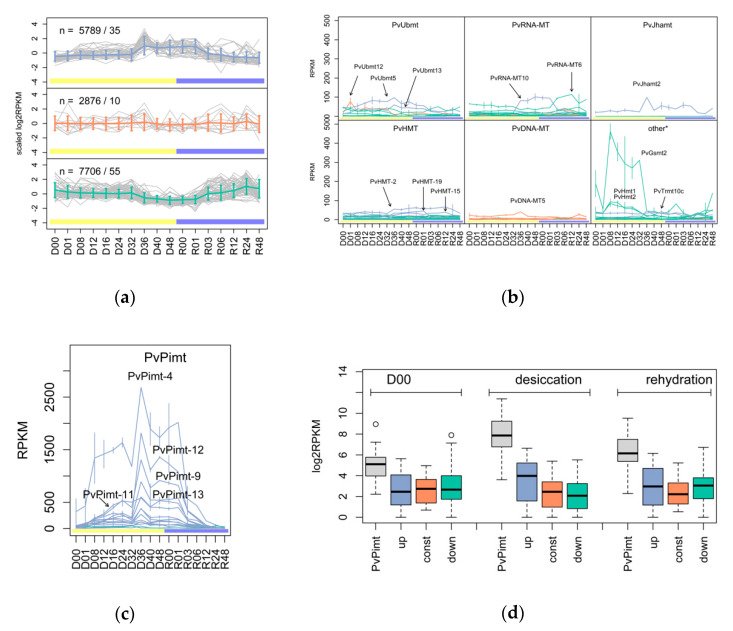
Expression profiling for defined SAM-dependent methyltransferases in desiccation (yellow)-rehydration (blue) experiments on *P. vanderplanki* larvae: (**a**) self-organizing maps (SOM) clustering of *P. vanderplanki MTases* into three groups, namely: upregulation (blue), stable (red), and downregulation trends (turquoise); *n* represents the number of genes/number of *MTases* in the group. (**b**,**c**) A view on the expression of different *MTases* by type. (**d**) Expression of *PvPimt* genes in comparison to other genes from defined clusters. D00—control, D01–D48—desiccation for 1–48 h (yellow bar), R00–R48—rehydration samples for 0–48 h (blue bar).

**Figure 2 insects-11-00634-f002:**
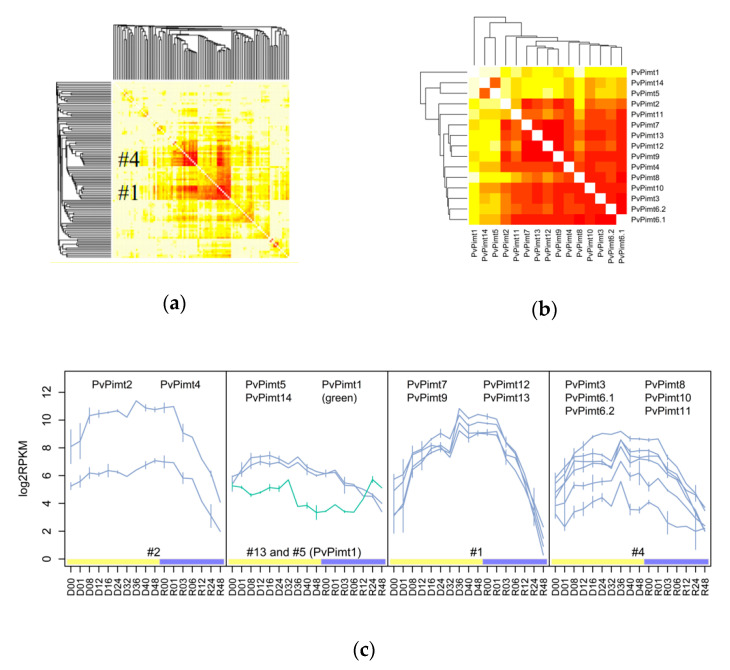
Co-expression network for *P. vanderplanki* SAM methyltransferases. (**a**) General view on the network, including all *Pv SAM MTases*. (**b**) *PvPimt* genes co-expression heatmap. (**c**) *PvPimt* expression profiles by WGCNA cluster. D00—control, D01–D48—desiccation for 1–48 h (yellow bar), R00–R48—rehydration samples for 0–48 h (blue bar).

**Figure 3 insects-11-00634-f003:**
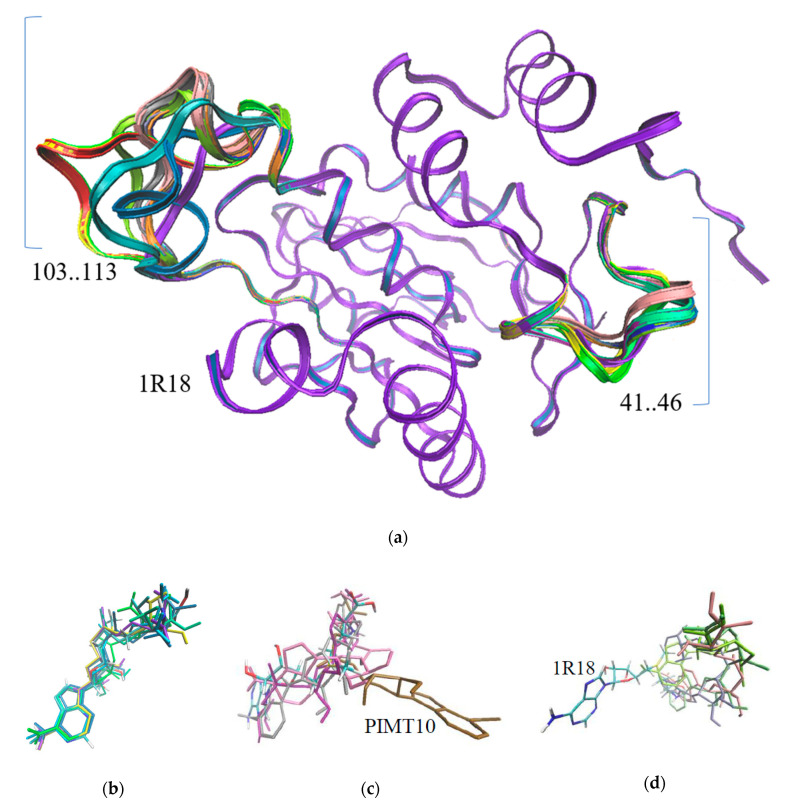
Modeling of PIMT spatial structures and docking with SAM; (**a**) predicted models of PIMTs using 1R18 as reference and position of SAM in different PIMTs of *P. vanderplanki*; (**b**) orientation of SAM similar to 1R18 in PIMT 1, 2, 5, 11, 12, and 14; (**c**) positions of SAM in PIMT 3, 6, 9, and 10; (**d**) different orientations of SAM in PIMT 4, 7, 8, and 13.

**Table 1 insects-11-00634-t001:** Discovered S-adenosyl-l-methionine (SAM) dependent MTases in *D. melanogaster (D. mel*) and chironomid species: *P. vanderplanki* (*P. van*), *P. nubifer* (*P. nub*), and *B. antarctica* (*B. ant*).

Insect Species	*D. mel*	*P. van*	*P. nub*	*B. ant*
Homocysteine *S*-methyltransferase EC 2.1.1.10 (Hmt)	2	2	1	1
Histone-arginine *N*-methyltransferase EC 2.1.1.125 (HMT)	10	6	8	5
DNA adenine-specific MTases EC 2.1.1.72	7	5	7	6
RNA MTases	31	24	26	18
Histone-lysine methyltransferase EC 2.1.1.43 (HMT)	18	21	18	19
PIMT EC 2.1.1.77	1	15	1	1
Diphthamide methyltransferase EC 2.1.1.98 (Dph)	1	1	1	1
Ubiquinone biosynthesis O-methyltransferases (Ubmt)	4	13	13	2
Protein-S-isoprenylcysteine O-methyltransferase EC 2.1.1.100 (Icmt)	1	1	1	1
Glycine/Sarcosine *N*-methyltransferase EC 2.1.1.156 (Gsmt)	1	2	1	1
Diphthamide methyltransferase EC 2.1.1.98 (Dph)	1	1	1	1
DNA (cytosine-C5) MTase EC 2.1.1.37	1	1	1	1
Juvenile hormone acid methyltransferase EC 2.1.1.325 (Jhamt)	1	4	4	2
Protein N-terminal methyltransferase EC 2.1.1.244 (NMT)	1	1	1	1
Spermidine/spermine synthases family EC 2.5.1.22 (Sps)	1	1	1	1
DNA (cytosine-C5) MTase EC 2.1.1.37	1	1	1	1
Leucine carboxyl methyltransferase EC 2.1.1.233 (Lcmt)	1	1	1	1
Methylene-fatty-acyl-phospholipid synthase EC 2.1.1.16 (Mfap)	-	1	1	1
Methyltransferase type 11 (Mt11)	3	2	2	2
Unchr_MeTrfase_Williams-Beuren Family (WBS_methylT)	1	1	1	1

**Table 2 insects-11-00634-t002:** The affinity energy of SAM and paralogous PIMT interactions.

PIMT	1	2	3	4	5	6	7	8	9	10	11	12	13	14	1R18
Energy, kcal/mol	−7.6	−8.9	−7.1	−7.8	−7.9	−7.7	−6.8	−7.1	−8.3	−7.0	−8.3	−8.3	−7.4	−8.1	−8.4

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
