# Peer review of "Diversity and Regulation of S-Adenosylmethionine Dependent Methyltransferases in the Anhydrobiotic Midge"

_insects, 2020, doi:10.3390/insects11090634_

Round 1

Reviewer 1 Report

Authors are trying various analyzes including in silico analysis, but what is the specific fit to Polypedilum vanderplanki by comparing these four species? I did not understand it at all. In discussion, the differences between the four types should be discussed firmly, or the points should be clarified by reviewing the resulting story.

The relationship between the gene clusters, the groups to which they belong, and various data are difficult to understand for the general readers. What does D00 to R48 mean? The time series results are described, but is it something that can be directly understood from the figure, not supplements? I would like you to speak clearly with the figures adopted in the main text or legends, not supplements.

Reviewer 2 Report

This study is a welcome addition to the field of anhydrobiosis and reports on the role of an interesting group of methyl-group transferring enzymes in anhydrobiosis. This understudied group of protein deserves more attention not only relating to their role in anhydrobiosis but in stress tolerance in general. However, I suggest some changes that might strengthen the paper. The introduction could be improved by providing some more information about PIMTs and their known contribution to enhancing lifespan and stress tolerance in other organisms. Furthermore, some more background could be provided about the functions of the other described methyltransferases and how these enzymes relate to each other. The results are providing interesting data on variety of different methyltransferases which have not been described in the introduction. Furthermore, the lack of activity of most PIMTs against NDB-DSIP could be substrate dependent or due to the transgenic system and I don’t think that the paper would be incomplete without these data. I may suggest a more thorough investigation of the enzymatic properties of recombinant PIMTs in a follow-up  study.

Minor:

The language needs some improvements. I would like to encourage the author to use shorter sentences with less clause insertions to reduce the potential for misunderstandings. Some examples included:

Lines 20-22: This sentence needs to be reworded.

Lines 34-38: This sentence needs to be majorly reconstructed.

Lines 51-52: This sentence is incomplete.

Furthermore, please make sure that all species names are italicized and make sure to add articles where needed (e.g. line 100: ‘The pPAL7 vector was used’ instead of  ‘pPAL7 vector was used’).

Reviewer 3 Report

This study examined the expression of methyltransferases in relation to dehydration in anhydrobiosis. There are some interesting points in these studies, but there are many aspects that needs corrected. 

  1. Spelling , grammar, and etc.  This needs substantial improvement with too many to point out that range from errors in the abstract to many issues with the references. 
  2. Line 23 - RNA-seq cannot predict differences in activity. 
  3. Line 27 - This does not mean these proteins lack the normal function.  Methylation could occur slightly differently or the conditions used to test the enzyme aren't physiologically accurate.  Specifically, some of these could only function at vary pH, salt concentrations, etc. This could be relevant during the course of dehydration.
  4. Line 41 - Please remove the info on PIMT genes from introduction unless a citation can be provided. 
  5. Details of WGCNA and SOM is lacking. These could not be replicated as described.  In addition, the lack of replicates can really skew these results.  What happens if the single replicate times are removed? Also, when I have tried to run WGCNA on less than 1000 genes output does not pass early quality control metrics.  
  6. Fig. 2b - How closely some of the gene change in expression indicates that co-mapping of reads might need to be examined.  Specifically, Cluster 1 is almost identical. 
  7. Modeling section - I'm not sure of the point of this section.  Does proteins structure match expression profiles? This need better integration with the rest of the paper. 
  8. Activity studies - how does this relate to dehydration? Two have activity, but how does this link to dehydration. 
  9. Line 220-230 - If the link between Lea and Pimt is so important, why is this not discussed in the results? 
  10. Line 263 - A lack of standard methylation is not proven.
  11. Discussion - This section need substantial revision to link the RNA-seq, gene prediction, modeling, and enzyme activity. 

Round 2

Reviewer 1 Report

It seems that an additional sentence as a response to comment 1 cannot be seen in the revised manuscript's discussion. If it is added, there will be no comment.

Author Response

Dear Reviewer,

I put this sentence into discussion part (230-235) in slightly different form, but same meaning

I forgot to mention the lines numbers in my previous reply, I'm sorry for this inconvenience

Best regards,

Ruslan